# Designing a Needs-Oriented Psychological Intervention for Chinese Women Undergoing an Abortion

**DOI:** 10.3390/ijerph20010782

**Published:** 2022-12-31

**Authors:** Na Wang, Debra K. Creedy, Mingna Zhang, Hong Lu, Elizabeth Elder, Jyai Allen, Li Guo, Qian Xiao, Jenny Gamble

**Affiliations:** 1School of Nursing, Capital Medical University, 10 Xitoutiao Road, Fengtai District, Beijing 100069, China; 2School of Nursing and Midwifery, Gold Coast Campus, Griffith University, Parklands Drive, Southport, Gold Coast, QLD 4215, Australia; 3School of Nursing and Midwifery, Logan Campus, Griffith University, University Drive, Meadowbrook, Brisbane, QLD 4131, Australia; 4School of Nursing, Peking University, 38 Xueyuan Road, Haidian District, Beijing 100191, China

**Keywords:** induced abortion, interviews, psychological stress, person-centered care, intervention development, China

## Abstract

Accessing good quality abortion care is a fundamental human right and contributes to achieving Sustainable Development Goals. However, well-designed abortion care that meets women’s needs is limited. This study aims to systematically develop an intervention to promote the psychological well-being of Chinese women undergoing an abortion. A five-step iterative approach informed by intervention mapping was undertaken to determine the intervention design. Step 1 used in-depth interviews with 14 Chinese women undergoing an abortion to assess real-life stressors and support needs. We identified eight stressors and found women’s support needs varied with the time trajectory of the abortion. Step 2 used a focus group discussion with care providers to select modifiable stressors that impact negative psychological outcomes. In Step 3 and Step 4, we determined and integrated the exact strategies to eliminate or mitigate possible modifiable stressors by incorporating information from in-depth interviews and the Transactional Model of Stress and Coping. The integrated strategies were instructional support, informational support, and timely communication. In Step 5, we composed the detailed intervention design according to the best available evidence and, to confirm content validity, consulted 10 women who had undergone abortion in the previous 2–6 weeks. The intervention was titled STress-And-coping suppoRT (START), which included four interacting components: (1) a face-to-face consultation at the first appointment; (2) a printed booklet with information on abortion, self-care, and managing emotions and intimate relationships; (3) a WeChat-based online public profile page offering the same information as the booklet; (4) a telephone hotline. This study paves the way for a new approach to addressing the psychological needs of women experiencing abortion in China. The rigorous process provides an example of developing tailored health promotion interventions.

## 1. Introduction

Globally, 73 million abortions occur each year, which equates to 30% of all pregnancies [1]. With the largest number of women in the world, China performs approximately 6–9 million induced abortions every year, corresponding to 15% of the world’s total numbers [2]. The Chinese government and public health agencies have launched a series of intervention programs, including post-abortion services, to reduce repeat abortions [3,4]. The current programs, however, focus on promoting contraception use and do not account for women’s emotional and psychological wellbeing.

Abortion is physically safe when conducted by trained healthcare providers. Severe physical complications following abortion are rare [5,6]. While denial of woman’s rights to access abortion is associated with adverse psychological outcomes [7,8], most women experiencing a first-trimester abortion also experience emotional and psychological distress [9,10]. The distress associated with abortion may be related to the circumstances of the unintended pregnancy and/or the process of care [9]. The environmental and personal factors that predict unintended pregnancy also predict mental health issues, no matter whether the pregnancy results in abortion or live birth [9,11]. These factors include conservative gendered social norms (valuing virginity, gender inequality, and/or stigmatization of abortion) [12], poverty, abuse [13], intimate partner violence [12,14], a history of emotional or mental issues [15], and personality factors [16,17] such as unconventionality, impulsivity, or an avoidance coping style. Importantly, if not addressed, these factors can place women at risk of unintended pregnancies, repeat abortions, and adverse mental health outcomes [9].

Unintended pregnancy and the abortion experience may increase a woman’s receptiveness to relevant health promotion programs [18]. Furthermore, accessing the abortion clinic provides health workers with the opportunity to identify and support vulnerable women [5]. Although the ideal way of screening high-risk populations should occur before any serious consequence, interventions after unintended pregnancies and abortions are still unquestionably profound in promoting women’s short- and long-term psychological wellbeing [15].

The research team conducted a systematic review to synthesize and integrate evidence surrounding the design and effectiveness of existing interventions in promoting the psychological wellbeing of women experiencing abortion [19]. The systematic review reported inconsistencies in the published literature with limited available high-quality evidence [19]. The review included 10 intervention studies (of which only one was conducted in China [20]), with the interventions classified into four categories: music therapy, information support, social support, and implementation of mandated waiting or counseling policies. None of the included studies discussed the intervention development process or had a theoretical basis, yet theory-based health interventions are more likely to be effective than non-theory-based health interventions [20]. The considerable heterogeneity and methodological limitations across studies contributed to conflicting outcomes, making it impossible to conclude which intervention optimized health outcomes [21]. The lack of high-quality and consistent evidence precluded us from adopting existing interventions or components to improve the psychological wellbeing of Chinese women undergoing abortion and led us to develop a tailored and targeted intervention.

The current paper outlines the development process of the STress-And-coping suppoRT (START) intervention following guidance for reporting intervention development studies in health research (GUIDED) checklist [22]. We developed the START intervention to address the gap in the literature and, more importantly, establish the relevance and feasibility of the intervention to help women in China cope with the abortion experience. The START intervention aims to: (i) alleviate depression symptoms; (ii) promote positive health behaviors; and (iii) improve women’s general experience of abortion.

## 2. Methodology and Methods

We followed the Medical Research Council (MRC) guidance for developing complex interventions [23]. The MRC guidance recommends the use of the best available evidence, appropriate theories, and pilot studies to address uncertainties in the intervention design [23].

In the current study, we used the Transactional Model of Stress and Coping (TSC) as a theoretical basis to underpin the START intervention design [24]. TSC has been widely used for understanding individuals’ coping processes in life events [25], including abortion [26]. Briefly, TSC considers individuals’ experience of a stressful event as a person–environment transaction process. When facing a stressor, an individual first evaluates the level of threat and then appraises their capacity to manage the stressor in light of available resources [24]. The results of this process influence an individual’s coping behaviors, which in turn affect outcomes, including psychological wellbeing, health behaviors, and functional status [24,25,26]. According to TSC, the impact of abortion on women’s wellness is mediated by women’s perception of abortion and the coping resources at their disposal [24]. Figure 1 illustrates the determinant framework of the START intervention based on the TSC.

We undertook a five-step iterative approach informed by the intervention mapping (IM) protocol for developing and evaluating health promotion interventions [27,28]. IM is widely used in composing psychological and behavior change interventions as its iterative pathway provides a blueprint for designing, implementing, and evaluating an intervention based on theoretical, empirical, and practical information [29]. The five steps and relevant tasks of IM for intervention development are: (i) conduct a needs assessment or problem analysis to identify what needs to be changed; (ii) create matrices of change objectives to identify which determinants should be intervention targets; (iii) select intervention strategies that match the determinants and translate these into practical applications; (iv) integrate the practical applications into an organized intervention program; and (v) engage intervention users to make sure of the appropriateness of the intervention design [29]. An overview of how we integrated the MRC guideline and the IM protocol and operationalized each step is illustrated in Figure 2. The following section describes the specific objectives and methods of each step.

### 2.1. Step 1: Assess Stressors and Needs

Abortion occurs in a socio-cultural context influenced by social norms and practices. As such, a sound understanding of the needs of Chinese women undergoing abortion was necessary to ensure the intervention addressed their challenges, support needs, and preferences. Objectives of this step were to (1) identify real-life stressors Chinese women face during the abortion process and (2) establish women’s needs and support preferences. We used Cohen’s definition of stressor, which is “demands made by the environment that upset individuals’ balance, affecting wellbeing, and requiring actions to restore equilibrium” [30].

We conducted a post hoc analysis of qualitative data derived from interviews with 14 Chinese women attending a hospital-based abortion clinic in Beijing, China. A convenience sampling strategy was used. All participants had received the procedure or medication to end a first trimester (<14 weeks) pregnancy without medical reasons. The interviews were conducted 2–4 weeks later, in Chinese, by the first author, to explore women’s experiences and perceptions of the abortion care received (see Appendix A for the English translation of original interview questions). The study was approved by the Peking University Institution Review Board (No. IRB00001052-16043, 2018). Written consent was obtained for the future use of de-identified data for scientific purposes.

In the current study, the dataset was re-analyzed with a focus on stress and coping. Deductive content analysis was used because existing data was re-analyzed from a new perspective [31]. Specific activities corresponding with the three-phase deductive content analysis approach were:Preparing: The dataset was reviewed by the first and fourth author for fitness for the current study based on a data-fitness assessment list for the second analysis of qualitative data (SAQD) guidelines [32] (see Appendix A for the assessment list of data fitness). After the two reviewers deemed the dataset was appropriate, an unstructured matrix informed by the aims of the study was developed by the first author (see Appendix A for the data analysis matrices);Organizing: The de-identified interview transcriptions were reviewed by the first and fourth authors (with detailed annotation) to gather relevant content according to the unstructured matrix. The content was grouped, and different categories were created within the matrix boundaries. Data abstraction was then undertaken. After the abstraction of each category, any discrepancies between reviewers were discussed with a third reviewer until a consensus was reached;Reporting: the first and the fourth author made a shared decision on how to model and report the results. To guarantee trustworthiness, details of the analysis process were recorded by all reviewers using a reflective memo. To preserve the original meaning of women’s statements, the preparing phase and organizing phase of data analysis was conducted in Chinese, and all three researchers involved were fluent Chinese speakers. The analysis matrices and reporting process were conducted bilingually to reduce potential bias and build a shared vision among the research team.

### 2.2. Step 2: Identify Modifiable Stressors

This step involved selecting possible modifiable stressors as targets to promote women’s psychological wellbeing. The research team and three service providers (two nurses and one doctor) reviewed and discussed the findings from Step 1 in relation to the following aspects: (a) which identified stressors need to be changed to improve women’s psychological wellbeing and (b) whether it would be possible to change the stressor in the short term with current resources. The discussion to select stress processes was facilitated by the first author.

### 2.3. Step 3: Specific Change Mechanism

Following the identification of modifiable stressors associated with women’s negative psychological experience of abortion, we sought to determine change mechanisms that could ameliorate or mitigate the stressors. A group meeting was held among the research team about the determinant framework of the START intervention. The basic premise was that if a strategy addressed or modified the determinants, it had the potential to ameliorate or mitigate relevant target stressors.

### 2.4. Step 4: Preliminary Design

Step 4 aimed to develop the preliminary design of the START intervention. The research team first reviewed all determined change mechanisms and grouped them into different intervention components based on the focus of the mechanisms. Then, guided by the determinant framework, we developed a logic model showing intervention mechanisms and what changes could be expected.

### 2.5. Step 5: Detailed Design

Step 5 focused on developing a detailed intervention design. Firstly, directed by the intervention description and replication (TIDieR) checklist [33], we developed an intervention delivery outline to address who, how, where, when, and how much each intervention component would be delivered. Secondly, we composed specific informational support content in accordance with the best available local and global evidence. Lastly, to ensure the content validity of this information, the research team invited 10 women who had experienced an abortion within the previous 2–6 weeks to review the content. Women were asked to rate each content theme according to (1) relevance—the extent to which the theme is pertinent to the stressor faced during the abortion process, (2) likely effectiveness—the possibility that the theme will successfully relieve stress levels, and (3) appropriateness—the extent to which the language use in the theme is appropriate [33]. Women were asked to rate each item using a four-point Likert scale from 1 = strongly disagree to 4 = strongly agree. If their rating was below the response of “agree”, they would be asked for suggestions for improvement [34]. Based on their suggestions, we revised the content.

## 3. Results

### 3.1. Step 1: Assess Stressors and Needs

Table 1 details participants’ age, parity, gestation age at abortion, abortion history, and interview characteristics, including the time point when the interview took place and duration.

Our analysis revealed specific stressors, unmet needs, and support preferences of Chinese women undergoing an abortion. We found women’s needs and support preferences varied with the time trajectory of the abortion, although several aspects were relatively constant (see Table 2). For example, women’s needs for emotional support, keeping the abortion a secret, and a non-prejudicial context that does not stigmatize abortions were evident. Women’s main stressors when initially presenting at an abortion clinic tended to be around resolving the unwanted pregnancy. These stressors related to time limitations, and/or reconciling the abortion decision with internal conflicts (e.g., moral beliefs on abortion vs. practical circumstances; what they want vs. what their partners or parents want; and current interests vs. plans for their future). At this stage, the need for unbiased and accurate information that supports and enables informed decision-making by the woman has been deemed a priority. Post-procedure, restoring normal emotional, functional, and social status in the context of having had an abortion was considered important. Confidential, integrated, and continuous support resources aimed at improving coping with negative emotional and psychological conditions, managing strained intimate relationships, and enhancing life skills to avoid similar problems were central to women’s reported needs. For most clients, privacy was also a priority concern. Services that threatened or were perceived to threaten privacy were unlikely to be used.

### 3.2. Step 2: Identify Modifiable Stressors

Among the eight stressors identified from Step 1, all were deemed as ‘modifiable’ to facilitate positive psychological outcomes. Two stressors, however, (‘difficultly accepting the decision with conflicting underlying beliefs’ and ‘societal stigmatization about abortion’) were deemed as not possible to change in a short time via clinic-based interventions. As a result, the remaining six stressors were identified as modifiable factors that the intervention needs to address.

### 3.3. Step 3: Specific Change Mechanism

The change mechanisms for each stressor are outlined in Table 3. They would be the guiding principles to direct the following decision-making on the START intervention design.

### 3.4. Step 4: Preliminary Design

Some of the determined change mechanisms overlapped or mapped onto each other and were aggregated into three intervention components: (1) instructional support, (2) informational support, and (3) timely communication channels. Furthermore, information needs were grouped into three themes and ten sub-themes (Figure 3). The logic model (Figure 3) also provided a clear description of what was included in each component of the START intervention as well as the process by which the different intervention components were expected to work to achieve the overall aim of the intervention.

### 3.5. Step 5: Detailed Design

A hierarchical outline for the delivery of each intervention component was developed (see Table 4). The instructional support would be offered via face-to-face consultations with a registered nurse after women make their abortion appointment. For informational support, the research team agreed on the necessity of both tangible and online information resources for women. A WeChat-based public profile page (also called the ‘official account’) was deemed appropriate for accessibility, confidentiality, and cost and time savings [35,36]. Timely communication channels included two methods: i) a hotline that women could call to speak to the research nurse during work hours and ii) a WeChat-based START online platform, in which women could message the research nurse.

Informational support content was developed according to the Family Planning—A Global Handbook for Providers (2018 edition) [5]; 2014 WHO Clinical Practice Handbook for Safe Abortion [6]; Abortion Care for women: a training toolkit [37]; and Exploring Abortion—A Collection of Self-Reflection and Sensitization Activities for Global Audiences [38]. A graphic designer designed the presentation of the START booklet and START online platform. The booklet was a 30-page, A5-size document comprising three themes and ten sub-themes (Figure 3). The START online platform had the same content as the booklet. This WeChat-based public profile page is accessible on a full range of devices and screen sizes (i.e., smartphones, desktops and laptops, and tablet devices).

Content validity of the booklet and platform was established with all ten women providing positive (agree or strongly agree) feedback about the “relevance” and “likely effectiveness” of each sub-theme. For the “appropriateness” of language, four women disagreed with one or more statements and provided suggestions for improvement. In total, five adjustments were made, involving two sections and three themes.

## 4. Discussion

This paper described the development of a needs-oriented psychological intervention for Chinese women undergoing an abortion. Following the IM protocol [28] we undertook a five-step iterative approach to gradually shape the intervention design. Using the TSC model as the determinant framework of intervention [24], we also presented the rationale behind the intervention design. This aligns with previous studies that reported health interventions informed by theories are more likely to be effective than those that are not [23]. The developed intervention (named START) is a complex intervention that combines multiple components that were delivered in various ways (i.e., a face-to-face consultation, a hard-copy booklet, a WeChat-based public profile page, and a hotline). Previous studies established that a multi-component approach and integration of technology are efficient ways to deliver mental health interventions [38,39]. For the above reasons, we believe it is reasonable to expect that START could generate a certain effect on women’s psychological health outcomes.

To our knowledge, this is the first study to explore Chinese women’s abortion experience from a stress and coping perspective using TSC as a theoretical basis. We explored Chinese women’s abortion experience and found that the stressors and support preferences of Chinese women vary with the time trajectory of abortion. This conclusion is consistent with research from other populations, including women accessing abortion services in the United States and the Netherlands [40,41]. The TSC model predicts that stress buffering is most effective when the type of support matches individuals’ priority needs in relation to the stressful event [30]. Thus, Chinese women’s priority needs at each stage of the abortion process were considered while developing the current intervention. For example, at the first abortion clinic appointment, a woman may think about whether to have an abortion or what type of abortion she should choose; therefore, offering information about birth control is not likely to be appropriate or may be less effective than giving the same information later in the process. We encourage future health workers or researchers to build a portrait of the specific stressors and needs of their target population to increase the potential acceptability and effectiveness of health services.

Using standardized approaches to developing and describing health service programs is important for achieving quality outcomes for service users [23,28]. The five-step iterative process informed by intervention mapping provides a strategy for health providers to know their clients and progressively determine the intervention design. As to the description of the intervention developing process, we followed the GUIDED checklist [22] to facilitate a complete, transparent, and consistent description of how the intervention was developed. We also used the TIDieR checklist to describe the intervention to promote the replicability of the intervention by other researchers and clinicians [33]. This study provides an exemplar for nurses and other health professionals in developing and reporting tailored health promotion interventions.

While the intervention was developed in a thorough and rigorous process, there are some limitations. Firstly, the stressors that the intervention targeted were identified from our interviews with women who had an abortion at a metropolitan hospital in China. The identified stressors may only represent those of women living in big cities, most of whom are well-educated and financially stable. Therefore, the developed intervention may not address all the stressors that jeopardize Chinese women’s psychological wellbeing when accessing abortion services. The specific components of the developed intervention should be interpreted cautiously when caring for women from other ethnic groups or even Chinese women from different backgrounds [42]. The other limitation is the intervention development process was conducted prior to the onset of the COVID-19 pandemic, which had a significant impact on health service practice and individuals’ daily life [43]. During the outbreak of the pandemic, many social activities were prohibited due to lockdown measures. This prevented women from accessing active coping strategies like exercise or seeking support from friends when facing an unintended pregnancy or abortion. In addition, the lockdown may make an already strained relationship between women and their partners worse [44,45]. With regard to accessing abortion services, women are required to have a COVID-19 test before making an appointment with the abortion clinic. These factors may alter women’s experience of abortion, as well as their support needs, and therefore the effectiveness and/or acceptance of the developed intervention during the pandemic may be affected.

Consistent with the IM protocol, the next step is to comprehensively assess the START intervention [28,29]. Best practice in intervention evaluation combines a variety of sources, ideally using different methods that do not share the same limitations to generate a full and comprehensive picture of the effectiveness, processes, and cost-effectiveness of an intervention for potential decision-makers [23]. Given the complex nature of the START intervention, the initial evaluation focuses on the feasibility and acceptability of the intervention [23,28]. The START intervention has undergone a pilot feasibility trial in a Chinese abortion clinic (ChiCTR2100046101) and confirmed the acceptability, feasibility, and preliminary effects of the intervention to promote psychological wellbeing among the targeted population in a real-world setting. Findings will be used in intervention refinement and future evaluations. Practical experience gained during the conduct of the clinical trial could also provide evidence for further implementation of the intervention. The evaluation of the START intervention will enrich the empirical literature on stress and coping, not only for the population of interest but for other researchers interested in the relationship between stress and individuals’ psychological responses.

## 5. Conclusions

We described a five-step iterative process to design a complex intervention targeting Chinese women undergoing an abortion to promote their psychological wellbeing. The process followed the IM protocol for planning health promotion interventions and integrated the TSC model as a theoretical basis. We also included insights from both women and abortion service providers from the outset to ensure that the intervention design is appropriate in the Chinese context. Our study illustrates how systematic development of a theory and needs-oriented intervention for a specific group is possible.

## Figures and Tables

**Figure 1 ijerph-20-00782-f001:**
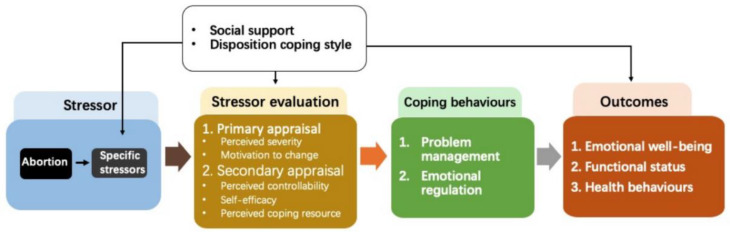
The determinant framework of the START intervention.

**Figure 2 ijerph-20-00782-f002:**
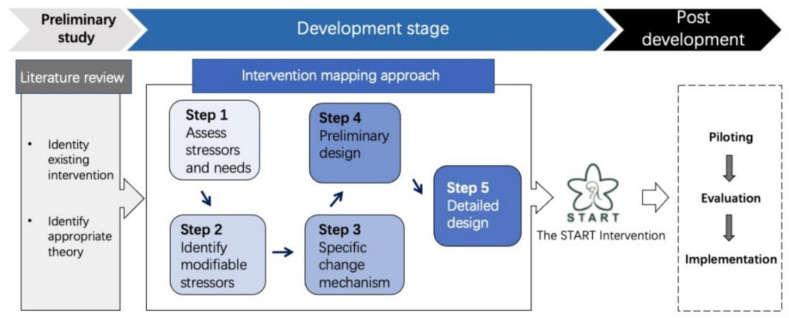
Flow diagram of the iterative development process [23,24].

**Figure 3 ijerph-20-00782-f003:**
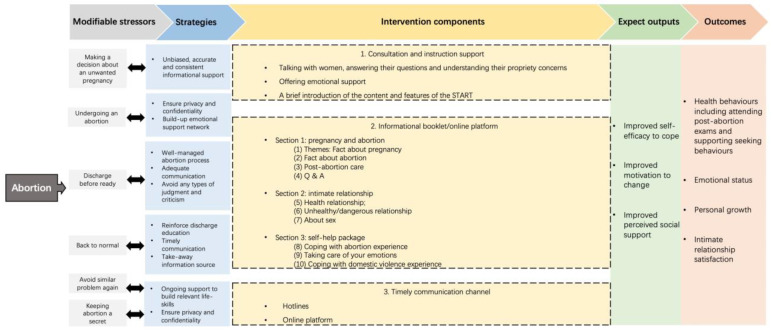
Logic model for the START intervention.

**Table 1 ijerph-20-00782-t001:** Demographic, reproductive, and interview characteristics of participants.

ID	Age	Gestation (Week^+Day^)	Number of Abortions	Interview Timepoint	Interview Length (min)
1	19	9^+3^	1	2 weeks post-abortion	46
2	21	11^+4^	1	2 weeks post-abortion	49
3	20	9^+6^	1	2 weeks post-abortion	55
4	25	9^+2^	2	6 weeks post-abortion	72
5	20	13^+1^	1	2 weeks post-abortion	33
6	24	10	1	2 weeks post-abortion	47
7	20	9^+4^	3	6 weeks post-abortion	52
8	18	10^+6^	1	2 weeks post-abortion	59
9	21	10^+2^	1	2 weeks post-abortion	40
10	24	12	2	6 weeks post-abortion	59
11	20	9^+4^	1	6 weeks post-abortion	48
12	18	10^+1^	1	2 weeks post-abortion	46
13	18	12^+4^	1	2 weeks post-abortion	35
14	29	11^+6^	2	6 weeks post-abortion	49

**Table 2 ijerph-20-00782-t002:** Stressors, needs, and support preferences of Chinese women undergoing abortion.

Time Frame	Stressors	Needs	Support Preferences
First appointment	Pressure deciding about an abortion Difficultly accepting the decision with conflicting beliefs underlying the decision	Unbiased, accurate, and consistent information support Emotional support and practical skills on how to build an emotional and social support network	Face-to-face consultation Easy-access online information platform
During abortion	Concerns about the abortion procedure (pain and bleeding symptoms) Possibility of being discharged before feeling ready	Safe, supportive, and well-managed abortion services Elaborate explanation about abortion practice Appropriate discharge education	Face-to-face communication inside the hospital Timely communication after discharge
Post-abortion	Desire to be “back to normal” in the context of having an abortion Possibility of a repeat unintended pregnancy or abortion	Ongoing support for coping with negative emotions and managing intimate relationships Help to develop life-skills to avoid similar situations	Confidential, interactive, and remote services
Whole process	Societal stigmatization in abortion Keeping the abortion a secret	Abortion-accepted sociocultural context Confidential and non-judgmental clinical environment	Confidential services

**Table 3 ijerph-20-00782-t003:** Change mechanisms for the START intervention.

Modifiable Stressors	General Determinants	Change Strategies/Mechanisms
Pressure deciding about an abortion	Perceived severity of the stressor Perceived ability to solve the stressor Perceived ability to manage emotional reactions Motivation to adopt preventive strategies Coping efforts Disposition coping styles Social support level	Unbiased, accurate, and consistent informational support Take-away information booklets (soft-copy or hard-copy)
Concerns about the abortion procedures (pain and bleeding symptoms)		Safe, normative, and well-managed abortion process Adequate communication Avoid all judgment and criticism Encourage the support person’s involvement
Possibility of being discharged before feeling ready		Reinforce discharge education Timely communication channel Take-away information resources
Desire to be “back to normal” in the context of having an abortion		Ongoing support
Possibility of a repeat unintended pregnancy or abortion		Develop life-skills to avoid similar situations Encouraging support person’s involvement
Keep the abortion a secret		Ensure privacy and confidentiality Build a reliable social/emotional support network Reassure women of the legal position of abortion

**Table 4 ijerph-20-00782-t004:** The START intervention delivery outline.

Components	Who	How	Where	When and How Much
Instructional Support	The first author	Face to face	Consultation room	After women made their abortion appointment; about 30 min
Informational Support	The first author	Booklet WeChat-based profile page	Consultation room WeChat	During Instructional Support; a free copy From the consultation until 6-weeks post- abortion; unlimited access till the closure of the project
Communication Channel	A registered nurse	Telephone WeChat-based profile page	Telephone WeChat	From the consultation until 6-weeks post-abortion; available during work hours

## Data Availability

The data involved in this study will be provided upon reasonable request to the corresponding author.

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
