# Peer review of "Designing a Needs-Oriented Psychological Intervention for Chinese Women Undergoing an Abortion"

_ijerph, 2022, doi:10.3390/ijerph20010782_

Round 1

Reviewer 1 Report (Previous Reviewer 2)

None

Author Response

We thank the reviewer for suggesting extensive English revisions. We have conducted a thorough review of the manuscript and have asked several native English-speaking academics to check the manuscript.

The following document details the editorial/grammatical corrections that have been made. 

Reviewer 2 Report (Previous Reviewer 1)

The authors have attended to all points raised in first review. Abstract, review of literature, methods adopted , results and discussion all meet the standards of good research approach. The iterative approach adopted in this qualitative study is unique and valid. The discussion is well balanced and reflects on pertinent points that are worthy to practitioners handling abortions . The tables and figures are well illustrated. References are complete. This is a good paper worthy of consideration for publication. I have reviewed the corrected version and find it good to go without any corrections.

Author Response

We thank the reviewer for the positive comments.

Reviewer 3 Report (New Reviewer)

This is a very thoughtful approach to evaluating stressors and mechanisms to alleviate them. I noted a couple typos.

I would very much like to see an equivalent evaluation of fetal mortality experiences, or other adverse pregnancy outcomes, e.g.. major birth defects, and associated stressors. To what extent are the stressors of pregnancy loss the same and to what extent do they differ. How can one evaluate whether an in depth qualitative study is generalizable to the "Chinese population". The authors wisely note that the stressors may well differ in different settings.

Author Response

We thank the reviewer for his/her constructive and carefully considered feedback which has strengthened this paper. 

This manuscript is a resubmission of an earlier submission. The following is a list of the peer review reports and author responses from that submission.

Round 1

Reviewer 1 Report

This qualitative research is based on a novel approach to determine common, yet unexplored factors that influence emotional and behavioral factors following abortions in a selected population in Beijing, China. 

The authors have reviewed current guidelines from WHO and have developed a framework to be the basis for conduct of this qualitative study. The approach to the design is based five iterative steps which appears a useful means in achieving the objectives. Agreeably, the transactional model of stress and coping has been employed and the incorporation of START including a Pilot study is useful. 

The authors have highlighted the factors that influence the four attributes that affects social support and disposition coping styles. The semi-structured questionnaire is acceptable. 

Overall the study-design and conduct of the study are acceptable. 

The figures included are illustrative and labelling is correct.

The authors have highlighted the strengths of the study which I agree. They have stated the limitations as subjects were from one institute in a large city . This statement would influence the conclusions drawn. 

Specific comments.

The abstract could include some positive findings. 

Paragraphs 4 & 5 in 'Introduction' are better included under 'Methods and Results'.

Some clarification about subjects is required: how were they selected? Was informed consent obtained? How long did each interview take place. State, in explicit terms, the language used in the interview . Demographic data including age, parity and previous abortions should be included under RESULTS. Kindly state the indication of abortions ( definition ) and the gestational age . Greater emotional issues arise when abortions are done after quickening and even after visualization of fetal activity when seen on ultrasonography. 

Reviewer 2 Report

1. Revise the manuscript title (Max. 12 words, including spacing)

2. Separate the "methods" and "results" 

3. Methodology should be detailed in the manuscript 

4. Expand the Discussion section 

Reviewer 3 Report

The authors described the development of a complex intervention for Chinese women undergoing abortion that aims to improve their psychological wellbeing. The intervention development process was grounded in the perspective and psychosocial context of the women who will receive it. Thanks for all the authors for this very good work.It is hoped that the results of clinical application can be reported in future studies.